# Real-Time Foreign Object and Production Status Detection of Tobacco Cabinets Based on Deep Learning

Chengyuan Wang, Junli Zhao *, Zengchen Yu, Shuxuan Xie, Xiaofei Ji and Zhibo Wan

College of Computer Science & Technology, Qingdao University, Qingdao 266071, China
* Correspondence: zhaojl@yeah.net; Tel.: +86-532-85953151

**Abstract:** Visual inspection plays an important role in industrial production and can detect product defects at the production stage to avoid major economic losses. Most factories mainly rely on manual inspection, resulting in low inspection efficiency, high costs, and potential safety hazards. A real-time production status and foreign object detection framework for smoke cabinets based on deep learning is proposed in this paper. Firstly, the tobacco cabinet is tested for foreign objects based on the YOLOX, and if there is a foreign object, all production activities will be immediately stopped to avoid safety and quality problems. Secondly, the production status of tobacco cabinet is judged to determine whether it is in the feeding state by the YOLOX position locating method and canny threshold method. If it is not in the feeding state, then the three states of empty, full, and material status of the tobacco cabinet conveyor belt are judged based on the ResNet-18 image classification network. Ultilizing our proposed method, the accuracy of foreign object detection, feeding state detection and the conveyor belt of tobacco cabinet state detection are 99.13%, 96.36% and 95.30%, respectively. The overall detection time was less than 1 s. The experimental results show the effectiveness of our method. It has important practical significance for the safety, well-being and efficient production of cigarette factories.

**Keywords:** deep learning; foreign object detection; production status detection; tobacco cabinet; cigarette factory



## 1. Introduction

In the tobacco production process, it is often necessary to confirm the status of the tobacco cabinet in real time. Tobacco factory workers need to climb up the tobacco cabinet for a long time and inhale a large amount of tobacco and dust particles every day, which seriously endangers their health. The aisles of tobacco cabinets are often multi-layered and narrow, which also creates huge safety hazards. After the production is completed, workers need to clean the interior of each tobacco cabinet. If the cleaning tool is forgotten in the tobacco cabinet, it will affect the quality of the next batch of cut tobacco, and may even cause the batch of tobacco to be scrapped. In order to reduce the cost of workers and eliminate the huge potential dangers of climbing the tobacco cabinet, the authors propose an automatic detection method that can detect the safety status and production status of tobacco cabinets in real time.

At present, most factories use visual inspection performed manually by workers, which often requires huge labor costs. Some factories identify and detect tobacco cabinets based on traditional machine learning methods, which have disadvantages such as low accuracy and long time consumption. In this paper, considering the powerful feature extraction ability of deep learning, the authors propose to realize real-time monitoring of tobacco cabinets based on the deep learning method. Compared with visual monitoring of workers and traditional machine learning methods, the accuracy and precision will be greatly improved.

Artificial Intelligence (AI) techniques such as Deep Learning (DL) and Machine Learning (ML) have been applied in many related fields related to object detection and image classification. With the development of object detection, the You Only Look Once (YOLO) series [1–6] always pursues the best trade-off between speed and accuracy in real-time applications. It plays an important role in facial recognition, industrial visual inspection and other fields. Zhang et al. [7] used the YOLOX [6] object detection network to count holly fruits and achieved a superior detection rate and excellent robustness in complex orchard scenarios. Hu et al. [8] improved the YOLOV4 [4] network to detect uneaten feed pellets in real time from underwater images. Compared with the past, the average precision is improved by 27.21%, and the amount of computation is reduced by approximately 30%. Yu et al. [9] improved the YOLOV4 network to detect face mask wearing. The results of the comparisons show that the accuracy of face mask recognition can reach 98.3%, and the frame rate is high at 54.57 FPS. Yang [10] proposed a face detection method based on YOLO [1]. It has stronger robustness and faster detection speed and can ensure higher detection accuracy in complex environments. It can be seen that YOLO object detection has the advantages of high recognition rate and fast recognition speed, and has been widely used in various visual detection tasks.

In recent years, object detection and classification have been applied to various fields. El-Sawy [11] uses the LeNet-5 network to recognize handwritten Arabic numerals, which brings significant improvements compared to machine learning classification algorithms. Zhang [12] improved the LeNet-5 convolutional neural network for pedestrian detection, which is better than the SA-Fast R-CNN algorithm. Li [13] improved the LeNet-5 network to analyze and classify the recognizable shapes in the urine sample image. It has wide applicability in the analysis of urine samples. Ma [14] proposed an effective smile detection method based on improved LeNet-5 and Support Vector Machine methods. The accuracy rates of the public MTFL database [15] and GENKI-4K database [16] reached 87.81% and 86.80%. He [17] et al. proposed Mask R-CNN, which extended Faster R-CNN by adding a branch for predicting an object mask in parallel with the existing branch for bounding box recognition. Instance segmentation, bounding-box object detection and person keypoint detection obtained good results. Mozaffari [18] et al. proposed an innovative convolutional module simulating the peripheral ability of human eyes. Significant results have been achieved in the field of semantic segmentation. Because the neural network can better extract features between different categories and calculate their differences, classification algorithms based on deep learning are often superior to traditional classification methods.

Real-time monitoring of tobacco cabinets is of great significance for tobacco production. Wang [19] proposed a cigarette category recognition method using image processing and deep learning technology to obtain the category and location information of cigarettes in the tobacco cabinet. Tobacco foreign body detection mostly adopted X-rayphotography and infrared detection methods [20], and a few of them adopted a hyperspectral imaging method [21]. However, due to the poor X-ray imaging and low contrast of the collected ray intensity, the sensitivity of this method is not high. Infrared detection uses the absorption of infrared light by the tobacco packet and foreign body to determine whether there is a foreign body. The range of foreign bodies that can be detected by infrared detection is limited. Due to the high cost of hyper-spectral imaging, its application to industry is still under discussion. Liu et al. [22] preprocessed tobacco image with a Laws operator to obtain its texture region and edge region. Subsequently, support vector machine (SVM) is used to subdivide the texture region and edge region to realize the simultaneous detection of multiple foreign bodies. As a tobacco image consists only of background, tobacco and foreign body, using the pattern recognition method for such a simple image is too cumbersome and less stable. Mi et al. [23] used a machine-vision-based direct detection scheme to detect abnormal objects in tobacco cabinets. The detection accuracy reached 97.8%, which has the advantages of high detection efficiency and low cost. However, Mi's detection method can only detect what kind of foreign object it is, and cannot locate the position of the foreign object. For other anomaly detection, Mozaffari et al. [24] proposed

using a one-dimensional CNN model for anomaly detection of Surface-Enhanced Raman Spectra applicable for Portable Raman Spectrometer in field investigations. In summary, a complete set of intelligent detection systems for tobacco cabinets is of great significance for tobacco production.

The rest of this paper is organized as follows. In Section 2, the materials and methods are introduced. The experimental and analysis results are described in Section 3 and the conclusion is given in Section 4.

## 2. Materials and Methods

### 2.1. Acquiring the Detection Data of the Tobacco Cabinet

This paper mainly focuses on the detection of foreign objects and the production status of the tobacco cabinet. The production status detection includes the feeding status detection of the feeder and the conveyor belt status detection of the tobacco cabinet.

For the collection of foreign object status, the authors define foreign objects as the objects that should not appear in the tobacco cabinet during the production of shredded tobacco, such as brooms, mops and people. Thus, the authors collect data on people, brooms, and mops in different tobacco cabinets. To avoid data duplication, the authors collect data in a variety of situations, such as workers, brooms and mops in different positions of the tobacco cabinet, etc. Since the light source in the workshop comes from the lamp, after the production is completed, the worker will turn off the light. At this time, the industrial camera will automatically switch to infrared mode to shoot gray images. Therefore, our data are divided into two types: gray images and color images with RGB three channels. We collected a total of 1548 foreign object data with an image size of 2560 × 1440.

As the cigarette factory is divided into different workshops such as the leaf cabinet, the expansion cabinet and the pre-distributed cabinet, the shape and size of the tobacco cabinets in different workshops are different, so the authors collect data on all kinds of tobacco cabinets. The authors use an industrial camera to collect 24 h of uninterrupted video recording of all the tobacco cabinets. To prevent the data from over-fitting, the authors only intercept one image of the same situation of the tobacco cabinet over a period of time. Finally, 684 empty material data, 802 full material data and 905 material data were collected.

Figures 1 and 2 show the original data of foreign object data and feeder in RGB and gray images. Figure 3 shows the status data of the tobacco cabinet conveyor belt.

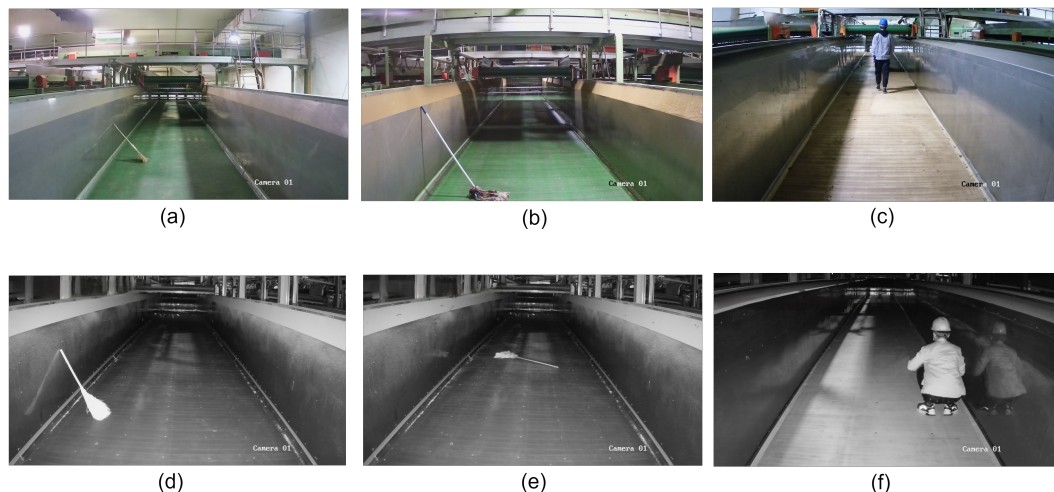

**Figure 1.** Foreign object data. (**a**) Broom (RGB image). (**b**) Mop (RGB image). (**c**) Human (RGB image). (**d**) Broom (gray image). (**e**) Mop (gray image). (**f**) Human (gray image).

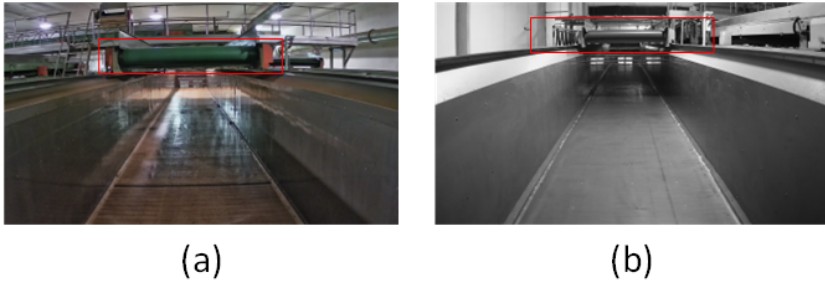

**Figure 2.** Feeder data. (**a**) Feeder for RGB images. (**b**) Feeder for gray images.

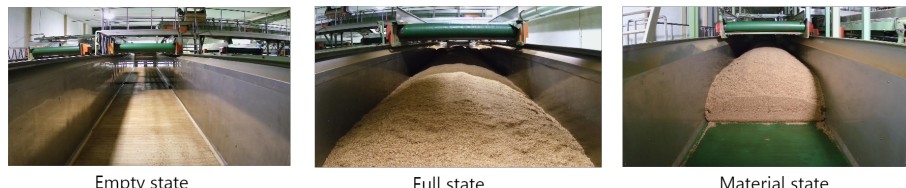

Empty state   Full state   Material state

**Figure 3.** Tobacco cabinet data. The left is the empty state, the middle is the full state, and the right is the material state.

## 2.2. Methods

According to the process of tobacco production, the authors design the pipeline of the production status and the foreign object detection of the tobacco cabinet as follows: firstly, the tobacco cabinet is tested for foreign objects based on the YOLOX [6], and if there is a foreign object, all production activities will be immediately stopped to avoid safety and quality problems. Secondly, it is judged whether the tobacco cabinet is in the feeding state. If it is not in the feeding state, then based on the ResNet-18 [25] image classification network in deep learning, the three states of empty, full and material status of tobacco on the tobacco cabinet conveyor belt are judged. The overall pipeline of our method is shown in Figure 4.

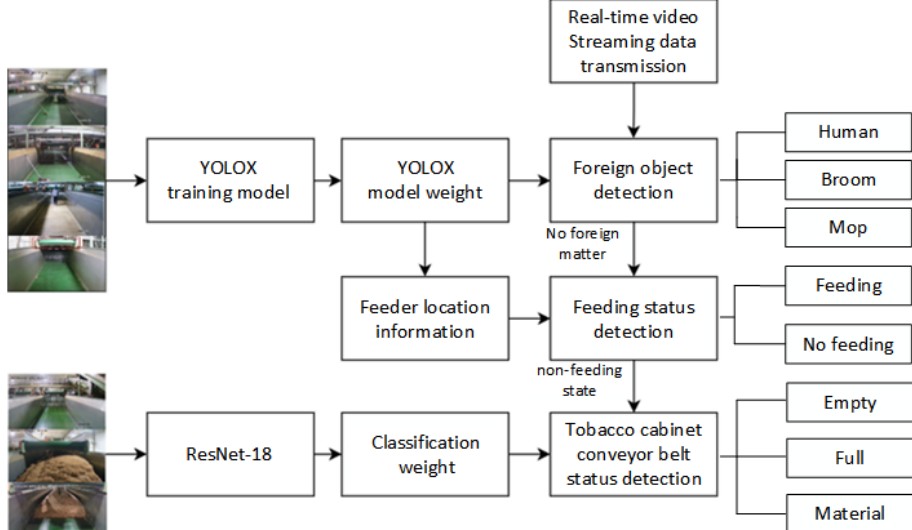

**Figure 4.** Overview of the proposed approach. The proposed aproach contains three steps: (1) The tobacco cabinet is tested for foreign objects based on the YOLOX. (2) It is judged whether the tobacco cabinet is in the feeding state. (3) If it is not in the feeding state, then based on the ResNet-18 image classification network in deep learning, the three states of empty, full and material status of tobacco on the tobacco cabinet conveyor belt are judged.

### 2.2.1. Foreign Object Detection of Tobacco Cabinet Based on YOLOX

Our foreign object detection is implemented based on YOLOX, which is essentially an object detection network in computer vision. YOLOX has greatly improved the detection accuracy and detection speed, which is of great significance for the industrial detection of tobacco cabinets.

The YOLOX [6] network is divided into three parts: Cross-Stage-Partial-connections Darknet (CSPDarknet [4]), Feature Pyramid Network (FPN [26]) and YOLO head (Figure 5). CSPDarknet is the backbone feature extraction network of YOLOX, and the input images are feature extracted in CSPDarknet. In the backbone part, three feature layers are obtained for the next step of network construction. FPN strengthens the feature extraction network. The three effective feature layers obtained in the backbone network will perform feature fusion in this part. The purpose of feature fusion is to combine the feature information of different scales. YOLO head is the classifier and regressor of YOLOX, which judges the feature points and determines whether the feature points have objects corresponding to them.

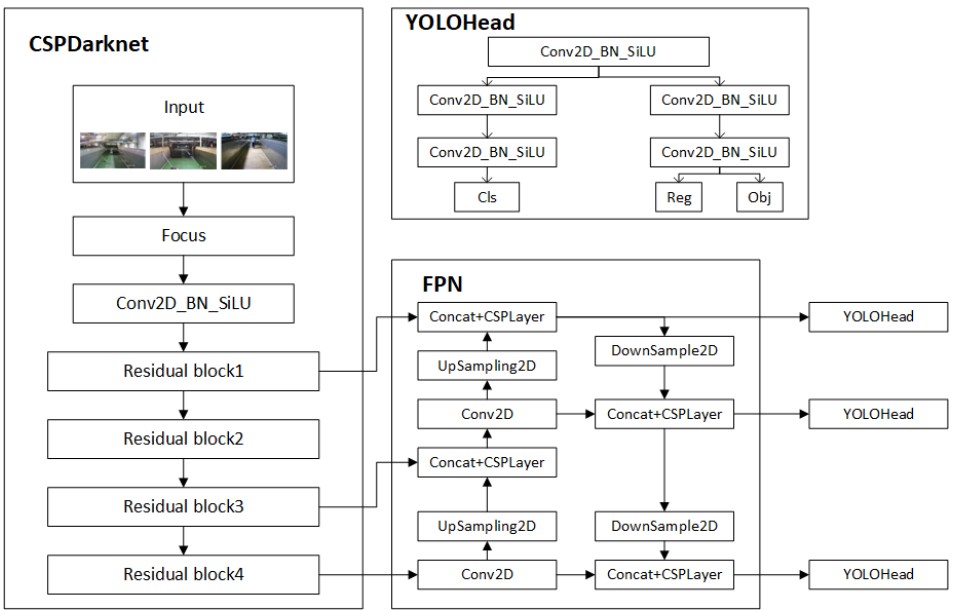

**Figure 5.** YOLOX network. It consists of three parts: CSPDarkNet, FPN, and YOLO head. CSP-Darknet is the backbone feature extraction network of YoloX; FPN is the enhanced feature extraction network of YoloX; Yolo Head is the classifier and regressor of YoloX.

A.    Backbone Network CSPDarknet

**Residual network.** We use the Residual network in CSPDarknet, and the residual convolution can be divided into two parts. The backbone part is a 1 × 1 convolution and a 3 × 3 convolution. Residual networks are characterized by being easy to optimize and capable of increasing accuracy by adding considerable depth. The internal residual block uses skip connections to alleviate the gradient disappearance problem caused by increasing depth in deep neural networks. In order to reduce the degree of overfitting, L2 regularization is performed every time of convolution. The calculation of L2 regularization is:

$$min : C_0 + \lambda ||\omega||_2^2 \tag{1}$$

$C_0$ is the loss function, $\lambda$ is the regular coefficient, $\omega$ is the weight vector and $||\omega||_2^2$ is the square root of the sum of squares of all weight values. After convolution is completed, Batch Normalization is performed. The standardized calculation of Batch Normalization is as follows:

$$y_i^b = BN(x_i)^{(b)} = \gamma(\frac{x_i^b - \mu(x_i)}{\sqrt{\sigma(x_i)^2 + \epsilon}}) + \beta \tag{2}$$

Among them, $x_i^b$ represents the value of the i-th input node of the layer when the b-th foreign object sample of the current batch is input, $x_i$ is the row vector formed by $[x_i^{(1)}, x_i^{(2)}, \ldots, x_i^{(m)}]$, m is the batch size and $\mu$ and $\sigma$ are the mean and standard deviation of the row. $\epsilon$ is used to prevent the extremely small amount introduced by dividing by zero; $\gamma$ and $\beta$ are the scales and shift parameters of the row.

**Cross Stage Partial Network.** The authors use the Cross Stage Partial Network (CSP-Net) structure. The original residual block is split into two parts: the main part is the same as the original operation; the other part is connected to the next layer of the network after a small amount of processing (Figure 6a).

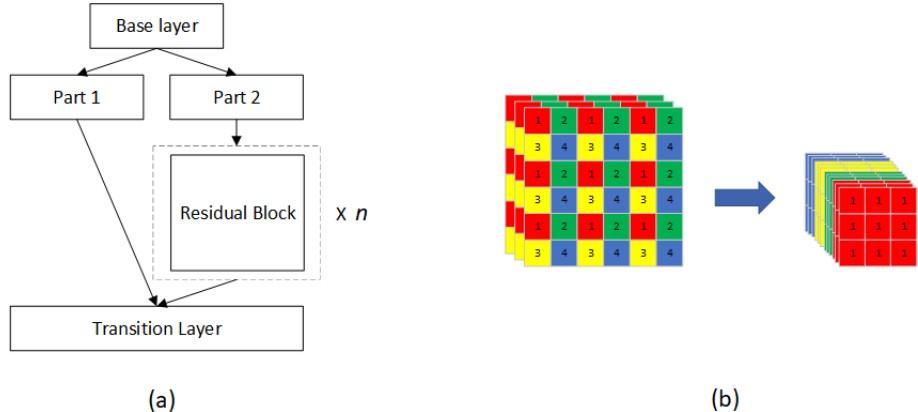

(a)                                                                                                    (b)

**Figure 6.** CSPNet and Focus network. (**a**) CSPNet: CSPNet is able to achieve richer gradient combinations while reducing the amount of computation. (**b**) Focus network: Using the focus layer can reduce parameter calculation and reduce the memory usage of cuda.

**Focus network [5].** We utilize the Focus network (Figure 6b) structure to sample a value every two pixels in an image. At this time, four independent feature layers are obtained, and then the four independent feature layers are stacked. The wide information is converted into channel information, and the input channel is expanded by a factor of four. The spliced feature layer becomes twelve channels compared to the original three channels.

**SiLU activation function [27].** For better smoothing, the SiLU activation function is adopted in our network. SiLU is smooth and non-monotonic. It can be seen as a smooth ReLU activation function and outperforms ReLU on deep models. SiLU function formula is:

$$f(x) = x \cdot Sigmoid(x) \tag{3}$$

**Spatial Pyramid Pooling (SPP) structure.** The SPP structure is applied to the backbone network, and feature extraction is performed through maximum pooling of different pooling kernel sizes to improve the receptive field of the network.

B.    Building FPN Feature Pyramid for Enhanced Feature Extraction

In the CSPdarknet network, YOLOX extracts three feature layers for object detection. The three feature layers are located in the middle layer, the middle–lower layer and the bottom layer in the backbone part. After obtaining the three feature layers, the authors use them to construct the FPN layer. FPN can fuse feature layers of different shapes, which is beneficial for extracting better features.

C.    Get Forecast Results with YOLO head

Using the FPN feature pyramid, the authors can obtain three enhanced features, and then the authors pass these three enhanced features into the YOLO head to obtain foreign

object detection prediction results. For each feature layer, the authors can obtain three prediction results:

**Reg(h, w, 4)** is used to judge the regression parameters of each tobacco cabinet feature point, and the prediction frame can be obtained after the regression parameters are adjusted.

**Obj(h, w, 1)** is used to judge whether each feature point contains an object.

**Cls(h, w, num_classes)** is used to determine the type of foreign objects contained in each feature point.

The three prediction results are stacked, and the result obtained by each tobacco cabinet feature layer is: Out(h,w,4+1+num_classses).

For foreign object detection, firstly, the data set marked with the foreign object is trained through the YOLOX network, and the training model file is generated. In the system, five frames of images are captured per second based on the video stream, and then the model file is used to detect whether there are foreign objects in the returned image. Finally, the predicted image and the predicted result are obtained. Figure 7 is an image of the result of foreign object prediction.

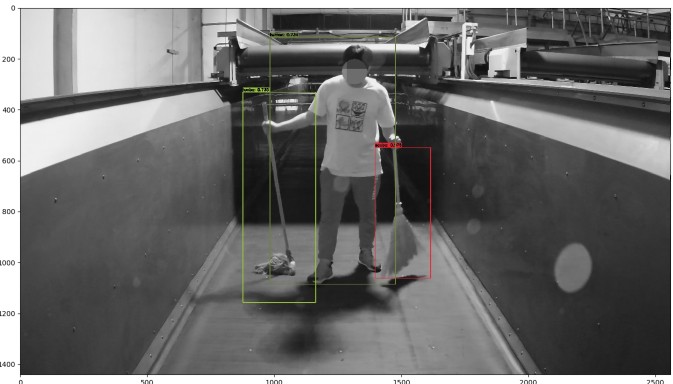

**Figure 7.** Foreign object detection result image. The left box detects a mop, the middle box detects people and the right box detects a broom.

### 2.2.2. Check the Feeding Status

If there is no foreign object in the tobacco cabinet, then we can check the feeding status. The YOLOX position locating method and canny threshold method is used to judge the feeding state. Firstly, capture five frames of images from the video stream every second, then use YOLOX to locate the target of the tobacco cabinet feeder on the images of the first and fifth frames and return the target position information. Set a threshold value of $T1$ to determine whether its location information has changed. Secondly, canny edge detection is used to extract the edge information of the first and fifth frame images. Perform an AND operation on the template of the feeder selected by the box to calculate the sum of the pixel values. Set Threshold $T2$ and compare to the sum of the pixel values in the template area to judge whether the feeder moves. If both conditions exceed the set thresholds, it is judged as the feeding state. The flow chart of the feeding state is shown in Figure 8.

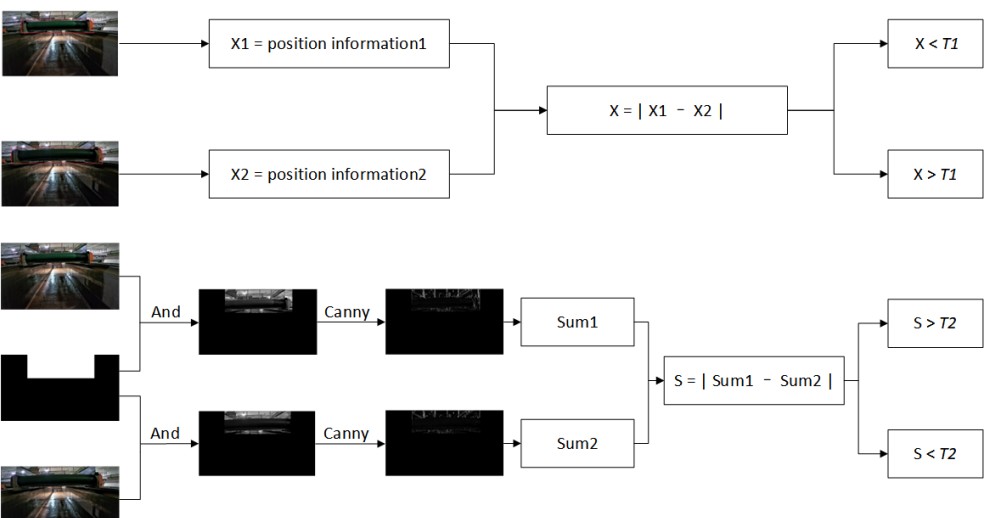

**Figure 8.** Feeding state detection. Firstly, YOLOX is used to determine whether the position of the feeder has moved, and secondly, the Canny threshold method is used to determine whether the template frame exceeds the set value. Only $X > T1$ and $S > T2$ are determined as the feeding status.

### 2.2.3. Tobacco Cabinet Conveyor Belt Status Detection

Tobacco cabinets have three states—empty, full and material state—during production and non-production. There are huge differences in the characteristics of the same state under different light sources, different workshops and different types of tobacco cabinets. This problem causes huge interference with traditional machine learning methods, such as edge detection and support vector machine classification. We use the deep learning image classification algorithm to classify the status of the tobacco cabinet, which can eliminate the influence of interference factors to the greatest extent.

In this paper, the ResNet-18 network is improved to identify and classify the status of the tobacco cabinet conveyor belt, and the input layer size is changed to $224 \times 224$. In order to reduce the number of parameters, the size of the convolution kernel is changed to a $3 \times 3$ small convolution kernel. The convolution layer and the first two fully connected layers use the ReLU activation function. The detection of the production status of tobacco cabinets is a three-classification problem, so the last fully connected layer uses the Softmax activation function.

## 3. Results and Discussion

We test our method on practical tobacco cabinets data from a tobacco factory in Qingdao. The experimental analysis will be illustrated from the following three aspects: foreign object detection, feeding status detection and tobacco status detection of the tobacco cabinet conveyor belt.

### 3.1. Analysis of Foreign Object State Detection Results

A total of 1548 images were collected for foreign object data, of which 770 were RGB image data and 778 were gray image data. Because the detection rate of humans is low in foreign object detection, the authors increase the amount of human data, collecting 728 human data, 459 mop data and 367 broom data. We take 70% of each part for training and 30% of the data for testing.

Our foreign object detection is based on YOLOX. Since the pretrained weights of the data are common to different datasets, we use transfer learning to train YOLOX. If the pre-training weights of transfer learning are not used, the weights will be too random, the feature extraction effect will not be obvious and the results of network training will be poor. We set the epoch to 200, use the SGD optimization function and set the batch_size to eight. In order to get a better training effect, we divided the training into a freezing phase and

a thawing phase. We set the first 50 epochs as the freezing phase. In the freezing phase, the model backbone is frozen, the feature extraction network does not change and only the network is fine-tuned. In the unfreezing phase, the model backbone is not frozen, the feature extraction network will be changed and all the parameters of the network will be changed at the same time.

To verify the accuracy of our method in foreign object detection, we compared it with YOLOV4 and YOLOV5. Figures 9 and 10 show the comparison of the loss and accuracy of YOLOV4, YOLOV5 and YOLOX under the same data, respectively.

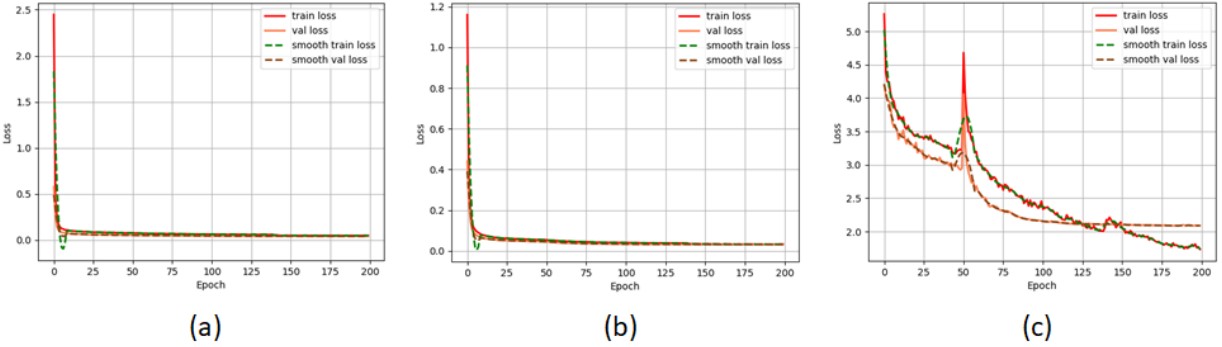

**Figure 9.** Comparison of training and testing loss. (**a**) YOLOV4; (**b**) YOLOV5; (**c**) YOLOX.

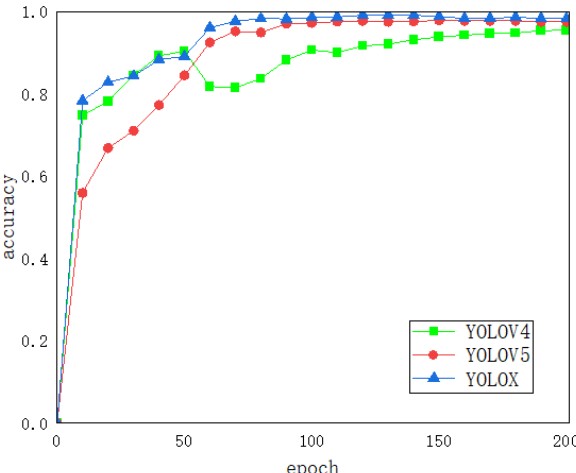

**Figure 10.** Test accuracy comparison.

Since YOLOX freezes the weights in the first 50 epochs, a noticeable change in loss can be seen at epoch 50. As can be seen from Figure 10, YOLOX has higher accuracy.

Different optimization functions have different effects on the training results. This paper uses the SGD optimization function. To verify the superiority of our method, we compare it with the Adam optimization function in YOLOX under other conditions being equal. Figure 11 shows the accuracy of SGD and Adam optimization functions in foreign object detection.

It can be seen from Figure 11 that the accuracy of SGD is better than that of Adam as a whole, and the highest accuracy rate in foreign object detection is 99.13%.

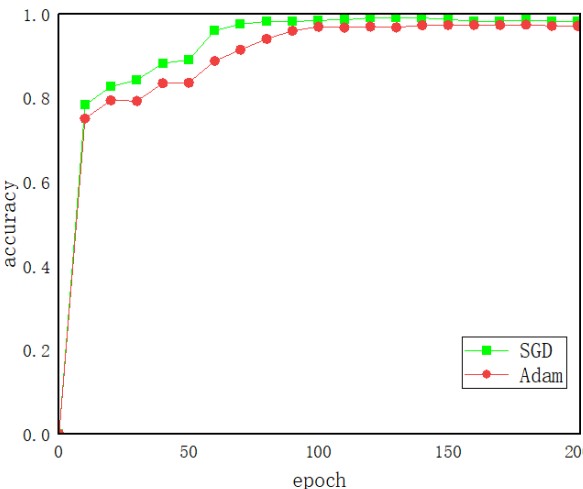

**Figure 11.** The accuracy of the SGD and Adam optimization functions in YOLOX in the test set.

*3.2. Analysis of the Test Results of the Feeding State*

We detect the feeding state of tobacco cabinets based on YOLOX position information and the Canny threshold method. When both the YOLOX position information and the Canny threshold method exceed the set threshold, it is determined to be in the feeding state. In other cases, it is judged as a non-feeding state. In order to verify the accuracy of the method in this paper, we conduct experiments in 55 different tobacco cabinets and conduct experiments on the following two cases: (1) use the YOLOX position information for detection; (2) use the YOLOX location information and Canny threshold method for simultaneous detection. Using the method in this paper, the highest accuracy rate in the detection of the feeding state is 96.36%. The experimental results are shown in Table 1.

**Table 1.** Accuracy comparison of two feeding detection methods.

| Method | Accuracy |
|:---:|:---:|
| Our method | 96.36% |
| The YOLOX position information | 94.55% |

*3.3. Analysis of the Test Results of the Production Status of the Tobacco Cabinet Conveyor Belt*

The production status detection of the tobacco cabinet conveyor belt is used to detect the three classification problems of empty, full and material states in the tobacco cabinet. We combined the improved ResNet-18 method for data training and collected 684 empty data, 802 full data and 905 data with materials. We train on 70% of the data and test on 30% of the data. The accuracy based on the ResNet-18 network reaches 95.3%.

When the Support Vector Machines (SVM) [28] method classifies the RGB image, because the color of the tobacco is yellow, and the color of the conveyor belt of some tobacco cabinets is also yellow, the classification effect of the SVM on the RGB image is disturbed. Our method is not affected by other interference factors to get better results. The highest accuracy rate is 95.3%. We also tested on LeNet-5 [29], AlexNet [30] and ResNet-101 [25] networks. The accuracy comparison results of the ResNet-18 and other methods are shown in Table 2.

**Table 2.** Comparison of the accuracy of ResNet-18 and other methods.

| Method | Number of Data Sets | Accuracy |
|---|---|---|
| ResNet-18 | 717 | 95.30% |
| LeNet-5 | 717 | 91.67% |
| SVM | 717 | 73.51% |
| AlexNet | 717 | 85.16% |
| ResNet-101 | 717 | 79.55% |

## 4. Conclusions

In the tobacco production process, the real-time monitoring of the tobacco cabinet is an important step in the production process, and the hidden dangers of tobacco products can be found in the production stage. In the past, tobacco cabinet inspection relied on manual visual inspection, which required huge labor costs. There is a safety hazard for workers who climb the smoke cabinet. This paper proposes an automatic tobacco cabinet production status and foreign object detection framework based on deep learning. The foreign object is detected based on the method of YOLOX object detection, the feeding state is detected based on the method of YOLOX location information and canny threshold detection and the state of the tobacco cabinet conveyor belt is classified into empty, full and material states using the ResNet-18 image classification method. The method in this paper achieves the effect of real-time automatic detection, and the accuracy is much higher than other machine learning methods. It greatly reduces the hidden safety hazards in cigarette factories through the original visual inspections by workers.

**Author Contributions:** Conceptualization, C.W. and J.Z.; Data curation, C.W. and X.J.; Formal analysis, J.Z. and S.X.; Investigation, C.W. and Z.Y.; Methodology, C.W.; Project administration, X.J. and Z.W.; Software, C.W.; Validation, J.Z.; Writing—original draft, C.W.; Writing—review & editing, J.Z. All authors have read and agreed to the published version of the manuscript.

**Funding:** This research was funded by the National Natural Science Foundation of China (No.62172247, 61702293, 61902203) and the National Statistical Science Research Project (No.2020LY100).

**Institutional Review Board Statement:** Not applicable.

**Informed Consent Statement:** Not applicable.

**Data Availability Statement:** Not applicable.

**Acknowledgments:** The authors are grateful to the anonymous reviewers for their careful checking of the details and for the helpful comments that improved this paper.

**Conflicts of Interest:** The authors declare no conflict of interest.

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
