# Peer review of "Real-Time Foreign Object and Production Status Detection of Tobacco Cabinets Based on Deep Learning"

_applsci, doi:10.3390/app122010347_

Round 1

Reviewer 1 Report

Dear Author(s),

The study is interesting. However, the writing style needs improvement. The quality of the papers needs improvement. Some of comments are the following to be addressed in the revised submission.

1. The abstract needs revision and rewriting to present what the article is about.

2. The literature should be included in details about the existing studies for tobacco-related real-time foreign and production status detection.

3. Whenever CNN or any statement without a reference is used, the reference must be included.For instance, YOLOX, LeNet-5, YOLOv4, Yolov5, MTFL database, GENKI-4k database, CSPDarknet, FPN, Focus network, Silu activation function, etc., do not have any references.

4. The figure labels and captions need rephrasing. They should include more details for the reader. For instance, in Figure-7, more details are provided.

5. Please avoid to include I, we, our usage, instead use, the authors or in this work/study and so on in the paper/work.It is recommended to avoid such nouns.

6. Discussion should be included to show the comparison of existing studies with the proposed method.

7. Wherever any comparison is presented, it is recommended to mention the environment. i.e., number of images, epochs, optimisers, etc.

8. It is recommended to use technical terms like images instead of pictures, datasets instead of databases, etc.

Thank you.

Good luck.

Reviewer 2 Report

The authors used the YOLOX object detection model for the detection of foreign objects and the production status of the tobacco cabinet. 

The article is an application of Deep learning for Industrial object detection. Although the contribution of the article is just an application, I mention here a few notes:

1- I don't see the rationale to use the LeNet network when there are many modern networks and more optimized ones available. Using the YOLO model makes sense when we need to find many many items in a scene with lower accuracy than other object detection techniques. Since YOLO is faster and has better performance speed. However, in this application, I think having higher accuracy is more important, and I can get why authors used YOLO versus other methods. Experimental results compare only YOLO models. 

2- Although visualization of all training/ validation trends is not a good idea, I see a sharp peak in training trends and un-normal behaviours in training and validation which doesn't explain clearly the reason for that.  

3- Why classification methods are old SVM? SVM is not a good candidate for comparison. I doubt the performance of ResNet-101, ResNet was the winner of the classification methods three years after the advent of LeNet. Even the improved version shouldn't perform those much better than the resnet model with 101 layers!! what is the reason? 

Reviewer 3 Report

The authors propose a method for object detection in tobacco cabinets using deep learning models.

Considering the state of the art, you mention a work about the same problem [16]. You should describe in more detail this work (disadvantages and advantages, methods, algorithms, etc.) and describe the improvements obtained with your approach.

You also consider an object detection problem. Why did you not consider it as a simpler classification problem with two classes (with and without foreign objects)? Do you need to know the kind of object? Does the process take different actions for different objects?

Your dataset is made of images of size 2560x1440. Do you run your models with these images or a resized version of them? How do you feed the model with both gray and RGB images?

What is the point of comparing YOLOv3, YOLOv4, and YOLOVx for the problem at hand (figure 10)? The relative quality of each of these modes is known. Also, you should consider a metric to compare the accuracy, loss, etc. of models. Checking these differences from the figures (in 10) is difficult and error-prone.

Also, what is the importance of comparing Adam against CGD? It is enough to state that SGD gave better results.

In table 2 you compare ResNet101 and others with improved LeNet5. Naturally, with the small dataset you considered for training, ResNet101 would probably give worse results. Given the low complexity of the problem, you should consider smaller good models, like ResNet18 with extra simplifications. Maybe it would achieve better results. There is no clear reason for the improved LeNet-5 to give better results since there are no improved layers in the model.

Performance is missing. You mention the system must analyze 5 images per second. Where are you running the algorithm? How fast is it? How do you feed the system with image data? Is it local processing or not?

You should compare the work in terms of accuracy and performance with previous works.

Round 2

Reviewer 2 Report

The newly submitted version is better than the previous one. On page 2, line 53, you are explaining object detection and classification. Also other places you are explaining deep learning in general terms. However, you forgot to talk about image segmentation. Models such as mask RCNN has the ability to be modified for the article problem instead of YOLO + ResNet. At least cite a few image segmentation methods, such as:

He, Kaiming, et al. "Mask r-CNN." Proceedings of the IEEE international conference on computer vision. 2017.

Mozaffari, M. Hamed, and Won-Sook Lee. "Semantic Segmentation with Peripheral Vision." International Symposium on Visual Computing. Springer, Cham, 2020.

Another method of Tabacco foreign body detection can be anomaly detection. for example for page 2, line 69, you can cite this:

Mozaffari, M. Hamed, and Li-Lin Tay. "Anomaly detection using 1D convolutional neural networks for surface enhanced raman scattering." SPIE Future Sensing Technologies. Vol. 11525. SPIE, 2020.

Reviewer 3 Report

The work is an application project and therefore does not introduce novelty. However, it has achieved good results for practical application and fits in the scope of the Applied Sciences journal.
